# Physiological and Transcriptomic Analyses of *Escherichia coli* Serotype O157:H7 in Response to Rhamnolipid Treatment

**DOI:** 10.3390/microorganisms11082112

**Published:** 2023-08-18

**Authors:** Shuo Yang, Lan Ma, Xiaoqing Xu, Qing Peng, Huiying Zhong, Yuxin Gong, Linbo Shi, Mengxin He, Bo Shi, Yu Qiao

**Affiliations:** Feed Research Institute, Chinese Academy of Agricultural Sciences, Beijing 100081, China; yangshuo01@caas.cn (S.Y.); malan0921@163.com (L.M.); xuxiaoqing@caas.cn (X.X.); pengqing@caas.cn (Q.P.); zhonghuiying173@163.com (H.Z.); 13704756082@163.com (Y.G.); linbo.shi@outlook.com (L.S.); hemengxin1998@163.com (M.H.)

**Keywords:** rhamnolipid, biosurfactant, transcriptome analysis, anti-biofilm, *Escherichia coli* O157:H7

## Abstract

Rhamnolipid (RL) can inhibit biofilm formation of *Escherichia coli* O157:H7, but the associated mechanism remains unknown. We here conducted comparative physiological and transcriptomic analyses of cultures treated with RL and untreated cultures to elucidate a potential mechanism by which RL may inhibit biofilm formation in *E. coli* O157:H7. Anti-biofilm assays showed that over 70% of the *E. coli* O157:H7 biofilm formation capacity was inhibited by treatment with 0.25–1 mg/mL of RL. Cellular-level physiological analysis revealed that a high concentration of RL significantly reduced outer membrane hydrophobicity. *E. coli* cell membrane integrity and permeability were also significantly affected by RL due to an increase in the release of lipopolysaccharide (LPS) from the cell membrane. Furthermore, transcriptomic profiling showed 2601 differentially expressed genes (1344 up-regulated and 1257 down-regulated) in cells treated with RL compared to untreated cells. Functional enrichment analysis indicated that RL treatment up-regulated biosynthetic genes responsible for LPS synthesis, outer membrane protein synthesis, and flagellar assembly, and down-regulated genes required for poly-N-acetyl-glucosamine biosynthesis and genes present in the locus of enterocyte effacement pathogenicity island. In summary, RL treatment inhibited *E. coli* O157:H7 biofilm formation by modifying key outer membrane surface properties and expression levels of adhesion genes.

## 1. Introduction

Enterohaemorrhagic *Escherichia coli* (EHEC) are zoonotic pathogens that can secrete Shiga toxins. These strains of *E. coli* are associated with ~20% of all foodborne illnesses and are associated with higher rates of hospitalization and fatality than other enteric pathogens such as *Salmonella* and *Campylobacter* [1,2,3]. The predominant EHEC serotype is *E. coli* O157:H7; this strain can cause severe disease in humans, including hemorrhagic colitis, hemolytic uremic syndrome, thrombotic thrombocytopenic purpura, and permanent organ failure [1,4]. *E. coli* O157:H7 is known to form biofilms in food manufacturing facilities, resulting in increased disinfectant resistance and contamination of food products [5].

Within biofilms, *E. coli* forms complex spatial structures through a dynamic, highly regulated process that is likely controlled by multiple factors, including both physicochemical and genetic components. The initial stage of biofilm formation is cell attachment to the substratum. This involves a number of physicochemical forces, such as hydrophobic and electrostatic attractions [6,7]. Regulatory pathways involved in *E. coli* biofilm formation include two-component signal systems (TCSs), quorum sensing (QS), and small noncoding RNAs (sRNAs). These pathways mediate bacterial chemotaxis, adhesion molecule function, activity of the secondary messenger cyclic di-GMP, and the bacterial stringent response, which results from environmental stress [8,9,10].

Preventing *E. coli* O157:H7 from forming biofilms is a widely used, effective strategy for reducing its health risks to humans [11]. Many natural and synthetic compounds have anti-biofilm activities that function through a variety of mechanisms to inhibit biofilm formation [9]. Amphipathic biosurfactants, which have both hydrophilic and hydrophobic moieties, can combat pathogenic bacterial biofilm formation by reducing surface tension and emulsifying immiscible phases [12,13]. Rhamnolipid (RL) is a well-studied biosurfactant that is produced by bacteria including *Pseudomonas* spp. and *Burkholderia* spp. This molecule contains one or two rhamnose moieties linked to one or two fatty acid alkyl chains [14]. RL is biocompatible and nontoxic to humans and can therefore effectively be used to prevent pathogen contamination in foods [15,16]. Although RL has not shown bactericidal activity against the Gram-negative bacterium *E. coli* due to host selectivity and the presence of the protective outer membrane, RL can inhibit *E. coli* biofilm formation and eradicate mature biofilms [17,18,19,20]. RL reportedly removes lipopolysaccharide (LPS) from the cell membrane and alters the surface hydrophobicity [17]. However, details of RL-associated changes and responses in *E. coli* O157:H7, such as host cell envelope permeability, regulatory network function, and metabolic activity, have yet to be clarified.

In the present study, we explored the effects of RL on *E. coli* O157:H7 biofilm formation, cell membrane integrity and permeability, and LPS release. High-throughput RNA sequencing (RNA-Seq) was also performed on *E. coli* O157:H7 cells within biofilms (RL-treated and control) to uncover the regulatory mechanisms responsible for the anti-biofilm activity. The results of this study serve as a strong basis for future studies seeking methods to control foodborne pathogens, particularly Gram-negative bacteria, using natural glycolipid biosurfactants.

## 2. Materials and Methods

### 2.1. Materials and Culture Conditions

The *E. coli* O157:H7 strain CICC 10907 was obtained from the China Center of Industrial Culture Collection (CICC). Bacteria were cultured in tryptic soy broth (TSB) (0.3% peptone, 1.7% tryptone, 0.25% K_2_HPO_4_, 0.25% glucose, and 0.5% NaCl) at 37 °C. The cell density of the culture (10^6^ colony-forming units (CFU)/mL) was adjusted to the appropriate concentration for each experiment using TSB. Commercial RL (a 3:1 mixture of Rha-Rha-C_10_-C_10_ and Rha-C_10_-C_10_, respectively) was purchased from Xi’an Ruijie Biotechnology Co., Ltd. (Xi’an, China). The RL was produced by *Pseudomonas aeruginosa*; it was 95% pure and had an average molecular weight of 614.3 g/mol. RL solutions were prepared in phosphate-buffered saline (PBS) with a pH of 7.4, then filter-sterilized.

### 2.2. Minimum Inhibitory Concentration (MIC) Determination and Biofilm Inhibition Assays

The RL MIC was determined using a standard microbroth dilution method as follows. RL was first two-fold serially diluted in sterile TSB to final volumes of 100 μL using 96-well microplates. *E. coli* O157:H7 culture (100 µL; 10^6^ CFU/mL) was then added to each well, bringing the final concentrations of RL to 0, 0.125, 0.25, 0.5, 1, 2, 4, 8, and 16 mg/mL. The plates were incubated at 37 °C for 24 h, during which time the optical density at 600 nm (OD_600_) was measured in each well every 6 h using a microplate reader (BioTek Instruments, Winooski, VT, USA).

The capacity of *E. coli* O157:H7 to form a biofilm on a polystyrene surface was determined using RL-treated and control cultures as described by Ma, Xu [6]. Films were examined with fluorescence microscopy on an LSM 700 confocal microscope (Zeiss, Jena, Germany) at 40× magnification. Quantitative biofilm measurements were conducted with thiazolyl blue tetrazolium bromide (MTT) assays as described by Xu, Peng [21].

### 2.3. Cell Membrane Integrity and Permeability Assays

RL solution was added to *E. coli* O157:H7 culture (10^6^ CFU/mL) to final concentrations of 0.125, 0.25, 0.5, and 1 mg/mL. After incubation at 37 °C for 24 h, each solution was centrifuged at 13,200× *g* for 10 min. The supernatant was removed, and the bacterial pellet was resuspended in PBS to an OD_600_ of 0.2. Flow cytometry was then conducted to determine the effects of RL on *E. coli* O157:H7 cell membrane integrity using the method described by Zhou, Wang [11].

*E. coli* O157:H7 outer membrane permeability was assessed using 1-N-phenylnapthylamine (NPN) as previously described by Efenberger-Szmechtyk, Nowak [22]. PBS was used as the negative control. Inner membrane permeability was measured with the protocol described by Ibrahim, Sugimoto [23], with PBS buffer and 1 mg/mL Polymyxin B serving as the negative and positive controls, respectively.

### 2.4. Cell Surface Hydrophobicity Assays

Assays to measure microbial adhesion to hydrocarbons were conducted with a modified version of the method described by Ma, Xu [6]. *E. coli* O157:H7 was cultured in 5 mL of TSB (containing 0.125, 0.25, 0.5, or 1 mg/mL RL) at 37 °C for 24 h. Each culture was centrifuged for 10 min at 13,200× *g*, then the bacterial pellet was resuspended in PBS buffer to an OD_600_ of 0.6 (5 × 10^7^ CFU/mL). The resulting *E. coli* O157:H7 suspensions (2.4 mL each) were mixed with 0.4 mL xylene for 3 min, then allowed to separate for 1 h. The OD_600_ of the aqueous phase was measured before and after xylene addition (OD_initial_ and OD_final_, respectively). Hydrophobicity was calculated as follows:Hydrophobicity (%) = (OD_initial_ − OD_final_)/OD_final_ × 100(1)

### 2.5. LPS Removal Assay

RL solution was added to *E. coli* O157:H7 cultures (10^8^ CFU/mL) to final concentrations of 0.125, 0.25, 0.5, 1, 2, 4, 8, and 16 mg/mL. After incubation at 37 °C for 24 h, each solution was centrifuged at 13,200× *g* for 10 min. The supernatant was collected and the amount of LPS removed from the cell membranes was quantified using the Purpald assay as described by Lee and Tsai [24]. The supernatant (50 μL) was first mixed with 50 μL of 32 mM NaIO_4_ and allowed to react for 25 min. Subsequently, it was incubated with 50 μL of 136 mM purpald reagent (in 2 N NaOH) for an additional 20 min, followed by a 20 min reaction with 50 μL of 64 mM NaIO_4_. The absorbance of the reaction mixture in each well was then measured at a wavelength of 550 nm using a plate reader.

### 2.6. Analysis of Adhesion-Related Gene Expression

#### 2.6.1. RNA-seq Experiment and Data Analysis

For RL-treated cells, 2 mL of 2.5 mg/mL RL solution was added to 18 mL of *E. coli* O157:H7 culture (10^6^ CFU/mL). For control cells, 2 mL PBS was added in place of RL. After incubation at 37 °C for 24 h, bacterial cells were harvested by centrifugation at 13,200× *g* and 4 °C for 20 min. The supernatant was removed, and the cells were washed with PBS to remove any culture residue. RNA extraction and cDNA library preparation were performed as described by Parkhomchuk, Borodina [25]. The libraries were sequenced on the Illumina NovaSeq platform to generate 150-bp paired-end reads. Low-quality reads were removed, and adapters trimmed with Fastp (v0.23.1). The R package ‘DESeq’ (v1.18.0) was used to identify significantly differentially expressed genes (DEGs) between RL-treated and untreated bacterial cells. The *p*-values calculated in ‘DESeq’ were adjusted using the Benjamini–Hochberg procedure to control the false discovery rate. The thresholds for DEG classification were adjusted *p*-value ≤ 0.05 and log_2_(fold change) ≥ 0.4. The identified DEGs were analyzed for functional enrichment of Gene Ontology (GO) terms using the ‘GOseq’ package in R (v1.18.0), which corrected for gene length bias [26]. Enrichment analysis was also conducted for the Kyoto Encyclopedia of Genes and Genomes (KEGG) biochemical pathways using KOBAS (v2.0) software [26,27]. GO terms and KEGG pathways were classified as significantly enriched where the corrected *p*-value was ≤ 0.05.

#### 2.6.2. Gene Expression Validation

Select genes in the RNA-seq dataset were validated with quantitative reverse transcription PCR (qRT-PCR). RNA was isolated from *E. coli* O157:H7 using a Total RNA Extraction Kit (Sangon Biotech, Shanghai, China) after treatment with 0.25 mg/mL RL. The isolated RNA was then reverse-transcribed into cDNA following the manufacturer’s instructions. Primers were designed for *lpxC*, *lpxH*, *fabA*, *accA*, *cyoA*, *cyoC*, *flgC*, *flgD*, *pgaA*, *pgaB*, and *dnaE* (Appendix A) using Primer Premier 5.0. Reactions were conducted in 2 × SG Fast qPCR Master Mix (High Rox; Thermo Fisher Scientific, Waltham, MA, USA) on the QuantStudio 3&5 Real-Time PCR System (Thermo Fisher Scientific). The thermocycling conditions included an initial denaturation at 95 °C for 3 min, followed by 45 cycles of 95 °C for 5 s and 60 °C for 30 s. Expression levels of each gene were normalized to the internal control gene *dnaE* using the 2^−ΔΔCt^ method [28]. qRT-PCR was conducted in technical triplicate.

### 2.7. Statistical Analysis

Statistical analyses were performed in SPSS Statistics v.21.0 for Windows (IBM Corp., Armonk, NY, USA). Statistically significant differences between samples were determined with one-way analysis of variance (ANOVA) and post-hoc Tukey’s test with a threshold of *p* ≤ 0.05. All measurements were conducted in triplicate.

## 3. Results and Discussion

### 3.1. Effects of RL on E. coli O157:H7 Growth and Biofilm Formation

The antibacterial activity of RL against *E. coli* O157:H7 was evaluated by examining bacterial growth after incubation with various concentrations of RL for 6, 12, and 24 h. RL showed significant (*p* ≤ 0.05) concentration-dependent effects on *E. coli* O157:H7 growth after incubation for 12 and 24 h (Figure 1A). However, even at the highest concentration (16 mg/mL), RL did not completely inhibit *E. coli* O157:H7 growth. These findings were consistent with those of a previous study by de Freitas Ferreira, Vieira [20], who reported that Gram-negative bacteria (such as *Salmonella enterica* and EHEC) are relatively resistant to RL, i.e., the MIC values are estimated at >2.5 mg/mL.

Biofilm formation capacity was next assessed using acridine orange. *E. coli* O157:H7 biofilms growing on polystyrene were stained with this dye, which is double-stranded DNA-intercalating; when bound, stimulation with blue light causes it to emit a green fluorescent signal [29]. In comparison with the negative control, 1 mg/mL RL significantly reduced *E. coli* O157:H7 biofilm formation (Figure 1B). The MTT method was then used to quantify the reduction in *E. coli* O157:H7 biofilm formation after RL treatment. This analysis showed that the inhibition of biofilm formation was concentration-dependent (Figure 1C). Specifically, treatment with 0.0625, 0.125, 0.25, 0.5, and 1 mg/mL of RL caused biomass reductions to 68.90 ± 7.01%, 41.37 ± 2.41%, 29.93 ± 0.77%, 28.48 ± 0.28%, and 27.68 ± 0.79%, respectively, of the untreated cell biomass. These findings were consistent with prior research showing that RL effectively inhibits biofilm formation by various bacterial and fungal species (including *Bacillus subtilis, Aeromonas hydrophila*, *P. aeruginosa*, *Staphylococcus aureus*, *Trichophyton rubrum*, and *Trichophyton mentagrophytes*) in a concentration-dependent manner [30,31,32].

### 3.2. Effects of RL on E. coli O157:H7 Cell Membrane Integrity and Permeability

*E. coli* O157:H7 cell membrane integrity was assessed using flow cytometry with the fluorescent probe propidium iodide (PI). When this method is used, plasma membrane damage increases fluorescence intensity, because PI can only enter cells with compromised membranes [33]. In the control group, only 3.54% of *E. coli* O157:H7 cells showed PI fluorescence (Figure 2A). RL treatment significantly increased the percentages of PI-permeable cells; among cultures treated with 0.125, 0.25, 0.5, and 1 mg/mL RL, 6.02%, 6.14%, 10.30%, and 13.20% of cells were fluorescent. The fluorescence intensities of *E. coli* O157:H7 cells also increased along with the RL concentration, indicating a higher degree of cell membrane rupture in response to higher RL concentrations. These findings align with previous studies of *B. subtilis* and *P. aeruginosa* showing that RL increases plasma membrane permeability and disrupts cell integrity [34].

We next analyzed changes in *E. coli* O157:H7 outer and inner membrane permeability as a result of RL exposure. NPN fluorescence was significantly increased (*p* ≤ 0.05) in bacteria treated with higher concentrations of RL (0.5–2 mg/mL) compared to the control, indicating that RL enhanced outer membrane permeability (Figure 2B). To evaluate the extent of RL penetration of the inner membrane, cytoplasmic β-galactosidase release was monitored among bacteria cultivated in a lactose-containing medium. β-Galactosidase is an enzyme that is located in the cytoplasm of bacterial cells and is involved in the metabolism of lactose. When the inner membrane of a bacterial cell is disrupted or damaged, β-galactosidase can leak out of the cell into the surrounding medium [35,36]. The results demonstrated that RL permeated the inner membrane in a dose-dependent manner at 0.125–1 mg/mL (Figure 2C).

### 3.3. Effects of RL on Cell Surface Hydrophobicity and LPS Release from the Outer Membrane

Cell surface hydrophobicity is an important factor in biofilm formation and can strongly influence bacterial capacities for surface adherence and complex community formation [37]. In general, bacterial cells with higher surface hydrophobicity exhibit a greater tendency to adhere to non-aqueous surfaces and form biofilms, potentially leading to persistent infections [38,39]. LPS is an essential component of Gram-negative bacteria such as *E. coli* O157:H7; it forms a complex network that stabilizes and maintains the integrity of the outer membrane. LPS removal can therefore significantly affect the *E. coli* outer membrane structure and function [17,40].

We here investigated the effects of RL on *E. coli* O157:H7 hydrophobicity and LPS removal. Compared to untreated cells, hydrophobicity did not significantly change in those treated with low RL concentrations (0.125–0.5 mg/mL) (Figure 3A). However, at a higher concentration (1 mg/mL), the cell surface hydrophobicity decreased significantly (*p* ≤ 0.05), from 39.16 ± 1.29% to 28.26 ± 1.44%. These findings were consistent with previous studies showing that high RL concentrations can reduce hydrophobicity on the cell surfaces of Gram-negative bacteria such as *P. aeruginosa* and *Sphingomonas* sp. Gy2b [41,42,43]. Furthermore, high RL concentrations (4, 8, and 16 mg/mL) led to LPS removal from *E. coli* O157:H7 membranes in a dose-dependent manner. This was comparable to a previous study by Bhattacharjee, Nusca [17], which found that high RL concentrations can trigger LPS release from *E. coli*. Such removal can enhance outer membrane permeability, potentially leading to bacterial cell lysis [44]. The finding that RL triggered LPS removal (Figure 3) was therefore consistent with the increased membrane permeability (Figure 2), and both of these factors explained the lower growth rates of *E. coli* O157:H7 treated with RL (Figure 1). Overall, these findings suggested that RL has the potential to modify *E. coli* O157:H7 cell surface properties and may therefore have important applications in controlling surface attachment and colonization by this pathogen.

### 3.4. Transcriptomic Responses of E. coli O157:H7 to RL Treatment

Transcriptomic analysis was conducted to investigate the mechanisms and signal transduction pathways that may be responsible for the observed reduction in *E. coli* O157:H7 biofilm formation after RL treatment. DEGs were identified in cultures treated with RL in comparison to untreated cultures. This analysis revealed 2601 DEGs, comprising 1344 up-regulated and 1257 down-regulated genes (Figure 4A,B). To determine potential biological mechanisms associated with these DEGs, GO functional annotation and enrichment analyses were performed. There were substantial differences in the enriched biological process (BP), cellular component (CC), and molecular function (MF) terms among the genes up-regulated in RL-treated compared to untreated cultures. The BP terms corresponding to the largest numbers of DEGs were metabolic and cellular functions associated with peptide and protein production. The most highly enriched CC terms were related to the non-membrane-bound cytoplasmic region of the ribosome. With respect to MF terms, the largest number of DEGs were annotated as structural components of the ribosome or RNA-binding molecules.

KEGG biochemical pathway enrichment analysis was also performed among the DEGs to identify crucial pathways involved in RL-mediated inhibition of *E. coli* O157:H7 biofilm formation. The majority of up-regulated DEGs were components of several key pathways: the ribosome, LPS, O-antigen nucleotide sugar, fatty acid, bacterial secretion system, and homologous recombination pathways. Conversely, down-regulated DEGs were primarily associated with microbial metabolism, glycolysis/gluconeogenesis and energy pathways, monosaccharide metabolism, and butanoate metabolism (Figure 4D, Appendix A). These results suggested that RL treatment caused significant alterations in the gene expression and metabolic pathways of *E. coli* O157:H7, leading to the inhibition of biofilm formation. Key genes related to these functions are discussed in more detail below.

#### 3.4.1. DEGs Related to the Cell Envelope

Gram-negative bacteria have a complex cell envelope that acts as a protective barrier against environmental stressors. The cell envelope is composed of two membranes (the outer and cytoplasmic membranes) and a periplasmic region containing the peptidoglycan (PG) layer. The inner leaflet of the outer membrane is composed of phospholipids and contains LPS [45]. As discussed above, treatment with the amphipathic biosurfactant RL removes LPS from the outer membrane of *E. coli*, resulting in improved cell permeability [17].

To determine how cells responded to the loss of LPS, we examined expression levels of LPS biosynthesis genes in RL-treated and untreated cultures. There were 29 LPS biosynthesis-related genes (eco00540) significantly up-regulated in RL-treated cells compared to the control (Appendix A). Up-regulated genes included those involved in the synthesis of three LPS domains (lipid A, 2-keto-3-deoxyoctonic acid [Kdo], and O-antigen) and in LPS structural modification. Specifically, 21 genes involved in Kdo2-lipid A and core LPS oligosaccharide biosynthesis were significantly up-regulated. Furthermore, 14 genes related to biosynthesis of nucleotide sugars for O-antigen, such as uridine diphosphate galactose, uridine diphosphate mannose, and guanosine diphosphate fucose, were up-regulated, as were 14 genes involved in PG biosynthesis. Genes encoding proteins present in the cell wall envelope, such as the outer membrane proteins OmpA, TolC, Murein lipoprotein (Lpp), and the Tol-Pal complex, were also significantly up-regulated (Table 1).

Fatty acids enable microbial cells to form essential membranous components that maintain cell integrity. We here found that 10 genes related to fatty acid synthesis (eco00061) were up-regulated in *E. coli* O157:H7 in response to RL treatment (Figure 4D, Appendix A). Treated cells also showed significant down-regulation of genes related to butanoate metabolism (eco00650); that metabolic pathway is responsible for maintaining the integrity and stability of the Gram-negative bacterial membrane [46] (Figure 4D, Appendix A).

*E. coli* uses envelope stress responses (ESRs) to monitor cell envelope integrity [47]. These factors can sense damage or defects in the cell membrane and activate envelope biogenesis. Overall, we observed that cell envelope biogenesis was up-regulated in *E. coli* O157:H7 in response to RL-induced cell membrane damage. We therefore hypothesized that these responses may have involved ESRs, as discussed below.

#### 3.4.2. DEGs Related to Bacterial Mobility

Bacterial chemotaxis is an adaptive process that enables bacteria to sense nutrient gradients and move toward favorable ecological niches; this helps them to avoid adverse environments [48]. Methyl receptor chemotactic proteins (MCPs) are responsible for sensing external stimuli and activating phosphorylation cascades, which ultimately lead to flagellar rotation. This allows bacteria to move away from stressful conditions [49]. Here, key genes related to the chemotactic signal transduction system, such as *cheA* and *cheY*, were significantly up-regulated in cultures treated with RL (Table 2).

The flagellum is an important adhesin that controls biofilm formation and participates in bacterial surface adhesion in *E. coli* [50]. Proteins encoded by *flg* and *fli* are responsible for biosynthesis and assembly of the flagellar basal body and hook, respectively; *motA* and *motB* encode stator proteins involved in flagellar rotation. Exposure to RL caused up-regulation of several *flg*, *fli*, *motA*, and *motB* genes, indicating an increase in flagellar assembly (Figure 5A; Table 2). This likely enhanced bacterial mobility and chemotaxis, suggesting that bacteria responded to RL by enhancing their capacity to move toward more favorable environments.

Some pathogenic bacteria contain a pathogenicity island called the locus of enterocyte effacement (LEE). LEE genes that encode the type III secretion system play significant roles in *E. coli* O157:H7 attachment and colonization of host cells [51,52]. Here, these genes were shown to be significantly affected by RL treatment. Specifically, the master virulence regulator gene *ler*, the chaperone gene *cesAB*, and the structural component genes *escR*, *escS*, *escT*, *escU*, *espA*, *espB*, and *espD* were all down-regulated (by 7.32-fold, 5.50-fold, 7.16-fold, 3.27-fold, 3.64-fold, 2.76-fold, 1.55-fold, 1.71-fold, and 1.81-fold, respectively) (Table 2). A previous study by Ma, Xu [6] showed that bacterial exopolysaccharide acts as an anti-biofilm compound; it up-regulates genes related to flagellar assembly and down-regulates LEE genes. Those findings are consistent with the results of the present study, suggesting that RL may be another anti-biofilm agent that can modulate LEE gene expression.

#### 3.4.3. DEGs Related to Energy Metabolism and Protein Synthesis

Cells require energy for growth, macromolecule biosynthesis (including synthesis of fatty acids, carbohydrates, and amino acids), and growth-associated maintenance in response to various conditions. Adenosine triphosphate (ATP), which is produced by pathways including glycolysis and oxidative phosphorylation, serves as the primary energy currency in living cells. Here, we found that the oxidative phosphorylation pathway (eco00190) was up-regulated in *E. coli* O157:H7 cells treated with RL. This included genes such as *nuoI*, *nuoJ*, *nuoL*, *nuoM*, *nuoN*, *sdhC*, *cyoE*, *cyoD*, and *cox10* (Figure 5B; Appendix A). RL treatment also resulted in significant up-regulation of the ribosome biosynthesis (eco03010) and aminoacyl-tRNA biosynthesis (eco00970) pathways; 77 and 18 up-regulated genes had these two annotations, respectively.

Genes associated with nascent protein transport, including 15 genes involved in protein export (eco03060) and 21 genes in the bacterial secretion system (eco03070), were significantly up-regulated among RL-treated cells (Figure 4D, Appendix A). Genes involved in protein export pathways (e.g., the Sec-dependent and twin-arginine translocation pathways), including *secA*, *secD*, *secE*, *secF*, *secG*, *secM*, *secY*, *yajC*, *yidC*, *ffh*, *tatC*, and *tatE*, were also up-regulated (Appendix A). These results suggested that RL exposure activated protein anabolism in *E. coli* O157:H7. Repairing damaged cell membranes is an energy-dependent process that requires RNA and protein synthesis; up-regulating the oxidative phosphorylation pathway provides ATP to support the repair process [53,54]. In summary, RL-induced membrane damage led to enhanced flux through energy metabolism and protein synthesis pathways to enable repair of the damage.

#### 3.4.4. DEGs Related to Stress Responses and Signal Transduction

Due to their additional outer membrane, which serves as a permeability barrier, Gram-negative bacteria have demonstrated greater resistance to antibacterial agents compared to Gram-positive bacteria [55,56]. The ESR in *E. coli* is complex and involves multiple stress-sensing mechanisms, such as the σE response, the Cpx response, and the *rcsBCDF* phosphorelay system, to monitor the cell envelope for damage and to restore homeostasis [47]. The *rcsBCDF* phosphorelay system is capable of sensing defects in the LPS layer of the outer membrane and of transducing stress signals across the cell envelope to the cytoplasm to activate expression of specific genes [57]. The Rcs stress response regulates capsule exopolysaccharide and flagellar synthesis [47,58,59]. *rpoS* acts as the effector of the Rcs response [47]; it positively regulates expression of genes associated with chemotaxis and flagellar biosynthesis while negatively regulating biofilm formation [60,61]. Consistent with those prior findings, we here observed that *rpoS* and genes in the Rcs pathway, including *rcsB*, *rcsC*, *rcsD*, and *rcsF*, were up-regulated in RL-treated cells (Table 3).

Cell envelope repair pathways are controlled by the σE response system, which enables RNA polymerase to increase transcription of genes involved in synthesis, assembly, and homeostasis of LPS and outer membrane porins [62,63]. The σE response promotes transcription of cell envelope biogenesis genes by releasing a σ factor (σE, encoded by *rpoE*) into the cytosol [63]. Here, RL-treated cells showed 1.35-fold, 1.33-fold, and 1.81-fold up-regulation of *rpoE*, *rseB*, and *degS* compared to untreated cells (Table 3). This suggested that RL treatment triggered the cell envelope repair pathway.

The Cpx response plays an important role in maintaining the cellular energy balance during envelope stress. It is activated by various conditions, such as misfolded inner membrane (IM) and periplasmic proteins, alterations in the lipid composition of the IM, and defects in the PG layer of the cell wall. This response pathway senses such stresses in the cell envelope, then relies on TCSs, namely *cpxA* (a sensory histidine kinase) and *cpxR* (a DNA-binding response regulator), to transduce the stress signal. The Cpx response can be regulated by nlpE when *E. coli* adheres to hydrophobic surfaces [47]. We here found that *cpxA* and *nlpE* were up-regulated by 1.41-fold and 1.68-fold, respectively, in RL-treated *E. coli* O157:H7 (Table 3). *cpxP* was previously believed to be a solely negative regulator of the Cpx response. However, recent discoveries have shown that *cpxP* also functions as a chaperone for clearing misfolded P pilus subunits and has direct interactions with *cpxA* [64,65]. Therefore, *cpxP* is now considered to have more significant regulatory and effector roles in the Cpx response than previously assumed [47]. In the present study, *cpxP* was shown to be up-regulated by 2.04-fold after exposure to RL (Table 3).

We also found that RL exposure up-regulated five genes in the nuo operon that encode components of NADH dehydrogenase I (NDH-I); five genes in the *cyo* operon that encode components of the cytochrome *bo_3_* complex; and one gene that encodes a component of the succinate dehydrogenase complex in *E. coli* O157:H7 (Figure 5B). These results were inconsistent with prior studies showing that the Cpx response down-regulates genes encoding the succinate dehydrogenase complex and aerobic respiratory complexes, such as NDH-I and cytochrome *bo_3_*, in *E. coli* [66,67,68]. It has previously been reported that some key pathways are shared between *E. coli* O157:H7 and other strains of *E. coli* but are differentially regulated. For example, bacterial chemotaxis and flagellar synthesis are positively regulated in *E. coli* O157:H7 under standard growth conditions, but negatively regulated in *E. coli* K12 [61]. Thus, the interactions between the Cpx stress response and respiratory complexes, such as NDH-I, cytochrome *bo_3_*, and succinate dehydrogenase, may show strain-specific regulatory differences that require further investigation.

Bacterial QS is a process that regulates cellular physiology, virulence, and biofilm formation through the use of signaling molecules called autoinducers (AIs) [69]. In *E. coli*, the most common AI that is involved in both intracellular and intercellular communication is AI-2, and the Lsr system is responsible for AI-2 detection, uptake, and signal transduction [70,71]. Deleting *lsrR*, *lsrK*, or *lsrB* interferes with QS signal transduction, resulting in reduced biofilm formation and bacterial mobility in avian pathogenic *E. coli* [72]. Here, RL treatment significantly impacted cellular QS signal transduction; *lsrR*, *lsrK*, and *lsrB* were down-regulated by 1.82-fold, 2.20-fold, and 3.31-fold, respectively, in response to RL treatment (Table 3). Furthermore, when *E. coli* is in the process of carbohydrate elimination, particularly when glucose is absent, adenylate cyclase (encoded by *cyaA*) is activated and catalyzes the creation of 3′5′-adenosine monophosphate (cAMP). cAMP then combines with a regulator, cAMP receptor protein (CRP), to stimulate Lsr expression [73,74]. RL exposure was here found to increase *cyaA* expression by 1.62-fold (Table 3). These findings suggested that RL treatment reduced AI-2 production and intracellular transport.

In *E. coli* O157:H7, several TCSs have been confirmed to participate in colonization and pathogenicity by sensing intestinal signals. One of these TCSs is BarA-UvrY, which is composed of the transmembrane sensor kinase BarA and the cytoplasmic response regulator UvrY [75]. The BarA-UvrY system activates expression of the *csrB* and *csrC* noncoding RNA systems in response to cellular stress. Subsequently, these two small RNAs bind to the RNA-binding protein CsrA, which regulates the expression of genes involved in biofilm formation, QS, and bacterial motility [76]. RL exposure here increased *uvrY* and *csrA* expression by 1.72-fold and 2.59-fold, respectively (Table 3). *uvrY* was previously shown to be a positive regulator of *E. coli* O157:H7 adhesion to cell surfaces and to activate LEE gene expression via Ler (an LEE master regulator) [77]. However, RL treatment was here associated with up-regulation of *uvrY*, but down-regulation of Ler regulatory and structural genes (Table 2). These results suggested that regulation of biofilm reduction and cell membrane repair are likely complicated processes involving multiple factors and genes. Further investigation will be required to determine the effects of *uvrY* on LEE gene expression in *E. coli* O157:H7 during RL treatment.

#### 3.4.5. qRT-PCR Validation of RNA-seq Data

To validate the RNA-Seq data, we selected 10 representative genes associated with biofilm formation and measured their expression levels with qRT-PCR. The genes (*lpxC*, *lpxH*, *fabA*, *accA*, *cyoA*, *cyoC*, *flgC*, *flgD*, *pgaA*, and *pgaB*) were selected because they were DEGs related to LPS and fatty acid biosynthesis, cytochrome *bo_3_* complex assembly, flagellar assembly, or polysaccharide intercellular adhesin. The expression levels as measured with qRT-PCR were highly consistent with the RNA-Seq data (Figure 6), validating the reliability of the sequencing-based analyses.

## 4. Conclusions

We here conducted a comprehensive study designed to determine how treatment with RL inhibits *E. coli* O157:H7 biofilm formation. The results indicated that RL inhibited biofilm formation by reducing outer-membrane hydrophobicity and triggering the release of LPS from the membrane; this decreased the cell membrane integrity and increased permeability. Gene expression analysis showed that RL treatment was associated with up-regulation of biosynthetic genes responsible for LPS synthesis, outer membrane protein synthesis, and flagellar assembly, and down-regulation of genes required for poly-N-acetyl-glucosamine (PNAG) and LEE biosynthesis. Based on the physiological and transcriptomic data, we propose a working model (Figure 7): RL exposure activated ESRs, including the σE, Cpx, and Rcs responses; bacterial QS; and the BarA-UvrY regulator. Activation of these systems led to the expression of genes involved in reducing biofilm formation and repairing membrane damage. Our findings demonstrate that RL modulated outer membrane surface properties and adhesion gene expression levels in *E. coli* O157:H7, resulting in the inhibition of biofilm formation. This study reveals a previously unknown mechanism of action for a promising anti-biofilm compound, potentially promoting future development of treatments for diseases caused by EHEC.

## Figures and Tables

**Figure 1 microorganisms-11-02112-f001:**
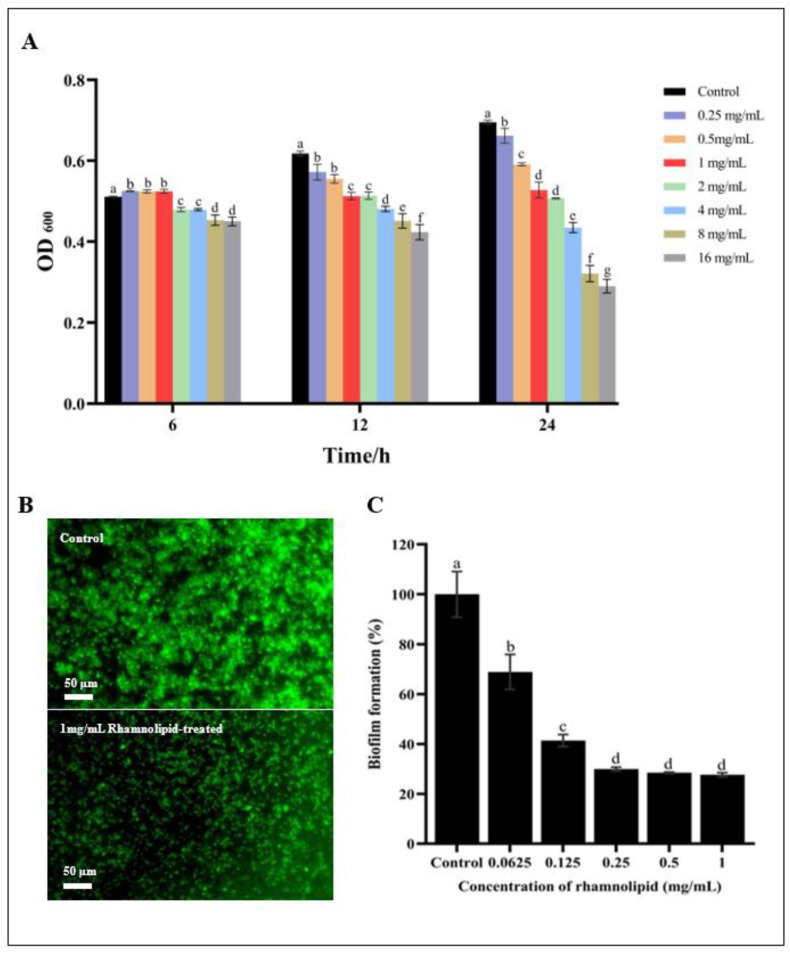
Effect of rhamnolipid on the growth (**A**) and biofilm formation (**B**,**C**) of *E. coli* O157:H7. Only the results obtained from each incubation period were compared for significant differences in (**A**). Different lowercase letters denote significantly different (*p* ≤ 0.05) OD_600_ values (**A**) and biofilm formation percentages (**C**).

**Figure 2 microorganisms-11-02112-f002:**
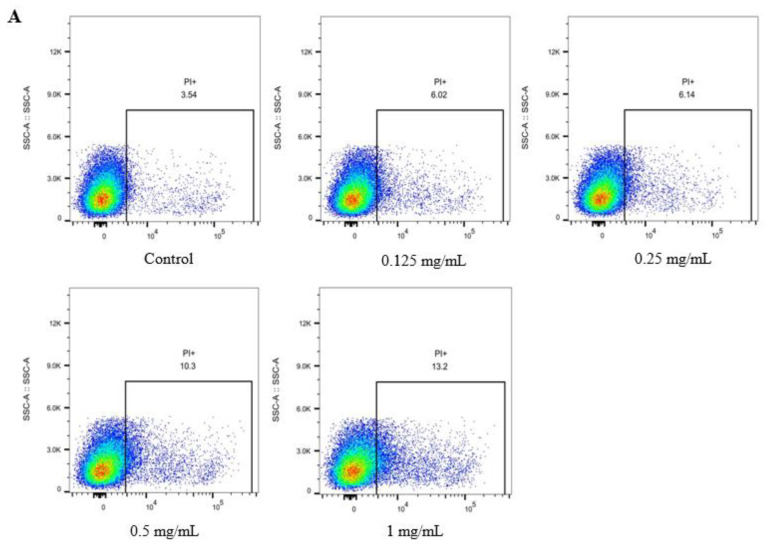
Effect of rhamnolipid on cell membrane integrity (**A**) and permeability (**B**,**C**) of *E. coli* O157:H7. Different lowercase letters denote significantly different (*p* ≤ 0.05) fluorescence intensity values (**B**) and absorbance values at 420 nm (**C**).

**Figure 3 microorganisms-11-02112-f003:**
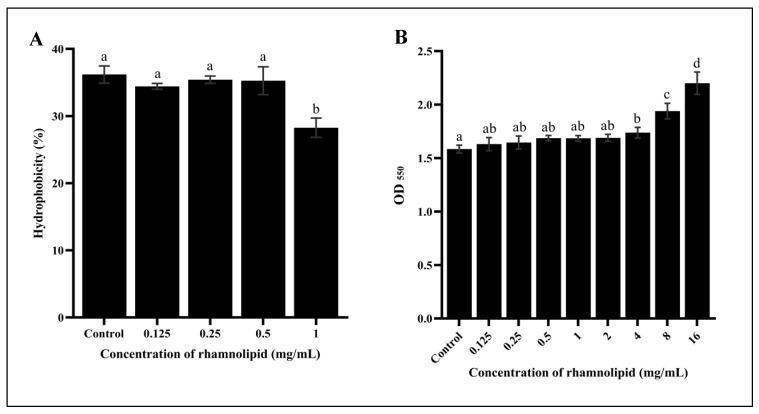
Effect of rhamnolipid on hydrophobicity (**A**) and the LPS release (**B**) of *E. coli* O157:H7. Different lowercase letters denote significantly different (*p* ≤ 0.05) hydrophobicity percentages (**A**) and OD_550_ values (**B**).

**Figure 4 microorganisms-11-02112-f004:**
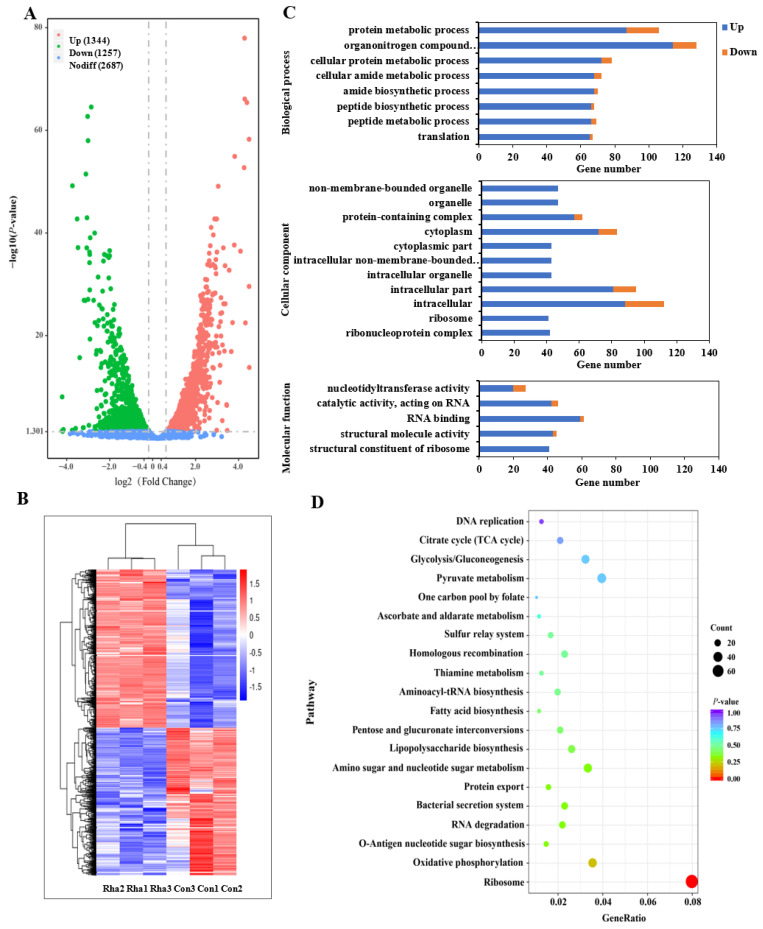
RNA-Seq gene expression profiles of rhamnolipid-treated *E. coli* O157:H7 (Rha) compared with the control group (Con). (**A**) The volcano plot of DEGs in bacterial cells (rhamnolipid vs. control). (**B**) The heatmap of DEGs. (**C**) GO enrichment analysis of DEGs annotated in three main categories: biological process, cellular component, and molecular function. (**D**) KEGG enrichment analysis of DEGs.

**Figure 5 microorganisms-11-02112-f005:**
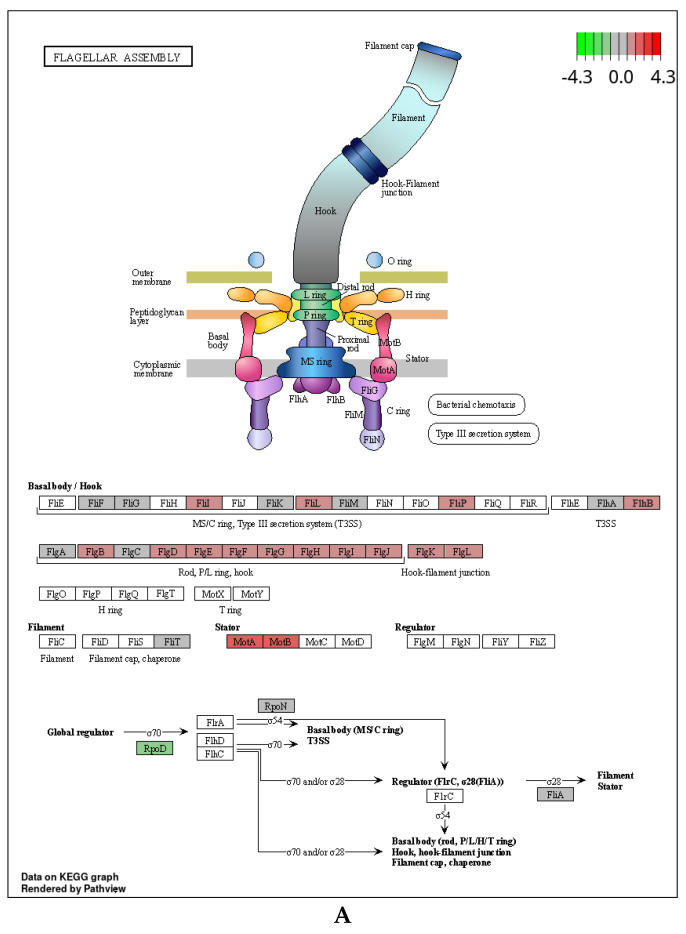
Transcriptional changes in genes involved in the significantly enriched pathway of *E. coli* O157:H7 in response to rhamnolipid. (**A**) The flagellar assembly pathway. (**B**) The oxidative phosphorylation pathway. Different colors indicate the value of log2 Fold Change of the differentially expressed genes (rhamnolipid vs. control).

**Figure 6 microorganisms-11-02112-f006:**
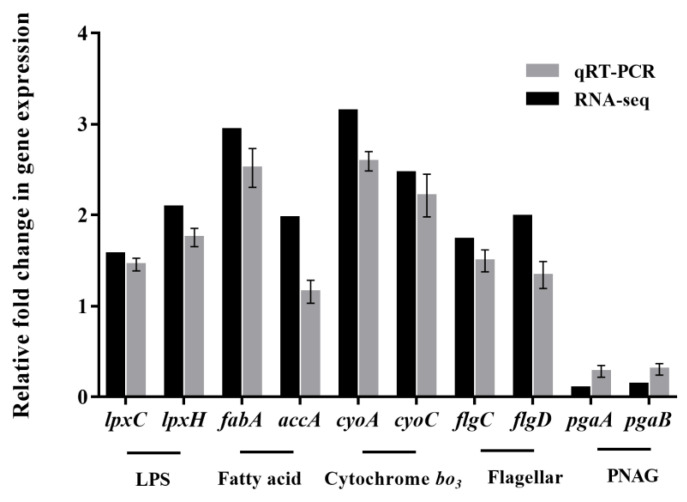
Relative change transcription of the selected genes determined by RNA-seq analysis and qRT-PCR in *E. coli* O157:H7 after treatment with rhamnolipid.

**Figure 7 microorganisms-11-02112-f007:**
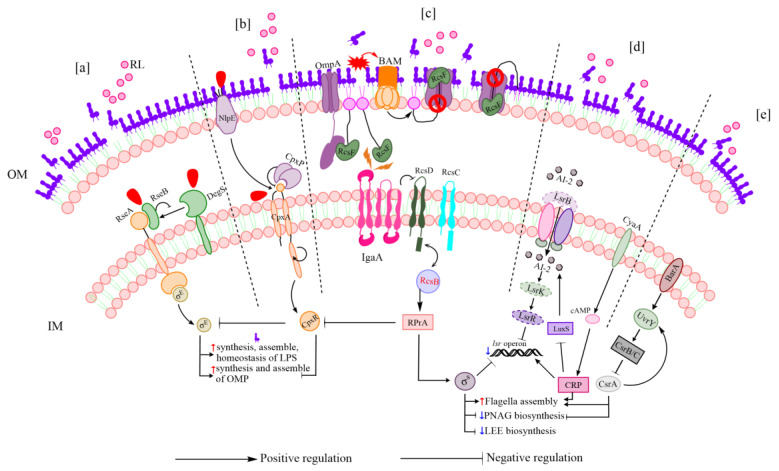
Anti-biofilm and membrane repair regulatory mechanisms of rhamnolipid (RL) against *E. coli* O157:H7. Schematic representation of the response to RL in *E. coli* O157: H7 involving the σ^E^ and Cpx responses, Rcs responses, bacterial quorum sensing, and *barA*/*uvrY* regulator. (**a**) σ^E^ response: After *rseB* is dislodged from *rseA* in reaction to LPS, *degS* senses the unfolded outer membrane proteins and triggered to cleave *rseA* to release σ^E^ to boost the expression of the target genes involved in cell envelope synthesis [47]. (**b**) Cpx response: By detecting signals from the cell envelope, NlpE activates *cpxA*, leading to the phosphorylation of *cpxR* and the modulation of gene expression [47]. (**c**) Rcs response: When cells experience envelope stress, the function of BAM in mediating the formation of *rcsF*/OMP complexes is impaired. As a result, BAM fails to assemble OmpC/F and bind to *rcsF*, causing free *rcsF* to accumulate in the periplasm and activate the Rcs response. This, in turn, strongly activates the *rprA* sRNA, which encourages the production of σs, the primary global regulator of biofilm development [56,59]. (**d**) Bacterial quorum sensing: Extracellular AI-2 molecules bind to the ABC transport protein *lsrB*, which then facilitates their transport via the *lsrACDB* transporter into the periplasm. In the periplasm, *lsrK* phosphorylates AI-2, which removes its repressor effects and stimulates the transport of *lsrACDBFG* [72,78,79]. In addition, the cAMP-CRP complex positively stimulates *lsr* operon expression, while σ^s^ negatively inhibits it [73]. The import of AI-2 within cells is decreased as a result of the down-regulation of this pathway. (**e**) *BarA*/*uvrY* regulator: After detecting cellular stress, the kinase *barA* catalyzes the transphosphorylation of the response regulator *uvrY*, which, in turn, triggers the transcription of the noncoding RNAs *csrB* and *csrC*. The interaction between the two small regulatory RNAs and the global regulator *csrA* controls the expression of genes involved in bacterial movement and biofilm formation [76]. LPS, lipopolysaccharide; BAM, β-barrel assembly machinery; OMP, Outer membrane protein; PNAG, Poly-N-acetyl-glucosamine; LEE, Locus of Enterocyte Effacement. Red and blue arrows indicate the up-regulation and down-regulation of genes, respectively.

**Table 1 microorganisms-11-02112-t001:** Major differentially expressed genes associated with cell wall envelope biosynthesis in *E. coli* O157:H7 treated with rhamnolipid.

Pathway	Function	Gene	Log_2_ (Fold Change)
Lipopolysaccharide biosynthesis	Kdo_2_-lipid A biosynthesis	*lpxA*	0.87
		*lpxB*	1.35
		*lpxC*	0.66
		*lpxD*	0.96
		*lpxH*	1.06
		*lpxK*	1.52
		*lpxM*	1.10
		*waaA*	1.32
	Core oligosaccharide biosynthesis	*kdsA*	0.86
		*kdsB*	0.46
		*kdsC*	0.64
		*kdsD*	0.39
		*waaG*	1.07
		*waaO*	0.97
		*gmhB*	0.79
		*waaC*	0.90
		*waaF*	0.73
		*waaD*	0.49
		*waaP*	0.79
		*waaQ*	0.92
		*waaY*	0.68
	O-antigen nucleotide sugar biosynthesis	*rffA*	0.67
		*rffC*	0.80
		*rffG*	0.81
		*rfbA*	0.65
		*wecB*	0.90
		*wecC*	0.95
		*cpsG*	2.06
		*galF*	1.25
		*galU*	0.68
		*gmd*	1.35
		*ugd*	1.86
	Lipopolysaccharide structural modification	*lpxP*	2.55
		*lpxT*	1.04
		*eptA*	0.98
		*eptB*	1.43
		*eptC*	0.86
		*arnT*	1.21
Peptidoglycan biosynthesis	Peptidoglycan synthase	*mrcA*	1.72
		*mrcB*	0.52
		*mrdA*	1.64
	Monomeric building block biosynthesis	*murA*	1.28
		*murB*	1.28
		*murC*	1.11
		*murG*	0.84
		*ddlA*	0.71
		*ddlB*	0.71
	Peptidoglycan biosynthesis	*dacA*	1.75
		*dacB*	1.12
		*dacD*	1.32
		*ispU*	1.97
		*ybjG*	0.63
Membrane protein	Outer membrane proteins	*ompA*	0.83
		*tolC*	0.64
		*pal*	0.64
	Lipoprotein in the outer membrane covalently crosslinked with peptidoglycan	*lpp*	1.03
	Tol-Pal complex	*tolA*	1.12
		*tolQ*	1.44
		*tolB*	0.79
		*ybgC*	1.59

Expression levels of all genes shown in the table are significantly different (adjusted *p* ≤ 0.05) between cultures treated with rhamnolipid and untreated samples. The log_2_(fold change) values represent gene expression in rhamnolipid-treated cells compared to untreated cells. Gene classifications and functions were taken from the KEGG website (Available online: https://www.genome.jp/kegg/pathway.html; accessed on 20 October 2022).

**Table 2 microorganisms-11-02112-t002:** Major differentially expressed genes associated with bacterial mobility in *E. coli* O157:H7 treated with rhamnolipid.

Pathway	Gene	Log_2_ (Fold Change)
Bacterial chemotaxis	*cheA*	1.51
	*cheB*	1.09
	*cheR*	0.90
	*cheY*	1.10
	*cheW*	1.12
Flagellar assembly	*flgB*	1.00
	*flgD*	0.99
	*flgE*	1.00
	*flgF*	1.42
	*flgG*	1.66
	*flgH*	1.43
	*flgI*	1.63
	*flgJ*	1.54
	*flgK*	1.42
	*flgL*	1.25
	*fliL*	1.13
	*flip*	1.13
	*flhB*	1.32
	*motA*	2.41
	*motB*	2.24
	*rpoD*	−1.24
locus for enterocyte effacement (LEE)	*ler*	−2.87
	*cesAB*	−2.46
	*escR*	−2.84
	*escS*	−1.71
	*escT*	−1.87
	*escU*	−1.46
	*espA*	−0.64
	*espB*	−0.77
	*espD*	−0.85

Expression levels of all genes shown in the table are significantly different (adjusted *p* ≤ 0.05) between cultures treated with rhamnolipid and untreated samples. The log_2_(fold change) values represent gene expression in rhamnolipid-treated cells compared to untreated cells. Biochemical pathway annotations were taken from the KEGG website (Available online: https://www.genome.jp/kegg/pathway.html; accessed on 20 October 2022).

**Table 3 microorganisms-11-02112-t003:** Major differentially expressed genes associated with stress responses and signal transduction in rhamnolipid-treated *E. coli* O157:H7 compared to untreated samples.

System or Response	Gene	Description	Log_2_ (Fold Change)	*p*-Adj
σ^E^ response	*rpoE*	RNA polymerase σ factor	0.46	0.04
	*rseB*	σ^E^ regulatory protein	0.41	0.04
	*degS*	Outer membrane stress sensor serine endopeptidase	0.86	3.0 × 10^−3^
Cpx response	*cpxA*	Envelope stress sensor histidine kinase	0.50	0.02
	*cpxP*	Cell-envelope stress modulator	1.03	4.0 × 10^−4^
	*nlpE*	Envelope stress response activation lipoprotein	0.75	1.0 × 10^−3^
Rcs response	*rcsB*	Transcriptional regulator	0.91	4.0 × 10^−4^
	*rcsC*	Two-component system sensor histidine kinase	0.83	6.0 × 10^−4^
	*rcsA*	Transcriptional regulator	0.98	0.01
	*rcsD*	Phosphotransferase	1.04	9.59 × 10^−5^
	*rcsF*	Rcs stress response system protein	0.88	2.0 × 10^−3^
	*igaA*	Intracellular growth attenuator protein	0.91	1.0 × 10^−3^
	*rpoS*	σ^S^, RNA polymerase σ factor	1.13	1.02 × 10^−5^
BarA-UvrY signal transduction	*uvrY*	UvrY family response regulator transcription factor	0.78	2.0 × 10^−3^
	*csrA*	Carbon storage regulator	1.38	2.75 × 10^−6^
Quorum sensing	*lsrB*	Autoinducer 2 ABC transporter substrate-binding protein	−0.86	0.02
	*lsrK*	Autoinducer-2 kinase	−1.14	5.31 × 10^−5^
	*lsrR*	Transcriptional regulator	−1.73	0.04

Expression levels of all genes shown in the table are significantly different (adjusted *p* ≤ 0.05) between cultures treated with rhamnolipid and untreated samples. The log_2_(fold change) values represent gene expression in rhamnolipid-treated cells compared to untreated cells. Annotations were taken from the KEGG website (Available online: https://www.genome.jp/kegg/pathway.html; accessed on 20 October 2022).

## Data Availability

Data are contained within the article or Appendix A.

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
