# Peer review of "Physiological and Transcriptomic Analyses of Escherichia coli Serotype O157:H7 in Response to Rhamnolipid Treatment"

_microorganisms, 2023, doi:10.3390/microorganisms11082112_

Round 1
Reviewer 1 Report
This is an excellent well conducted and written study, with an original contribution to the knowledge of first report of Physiological and Transcriptomic Analyses of Escherichia Coli Serotype O157:H7 in Response to Rhamnolipid Treatment.
I encourage it’s acceptance after appropriate minor modifications as outlined below:
L31: For a better understanding of these phenomenon and also to ensure an complete and objective overview, I recommend you to cite a few more scientific articles (e.g. Gligor et al., 2022, MICROBIOLOGICAL AIR POLLUTION IN AREAS WHERE PIGEON COLONIES LIVE, IN TIMISOARA, ROMANIA, https://agmv.ro/wp-content/uploads/2022/09/05_12_Gligor_8-c_compressed.pdf).
Reviewer 2 Report
The main question addressed by this article is the impact of rhamnolipid on E.coli disinfection. The paper is original and compared to other published material, it adds the analysis of gene expression.
I have a few comments regarding the methodology of the study:
Line 38: specify with which strains E. coli forms multispecies biofilms
Line 78: Specify what was the initial concentration of cells. This is important, since this impact the results.
Section 2.2 again, information on cell concentration is missing. This is vital to interpret results correctly.
Section Methods: For all experiments information on parallels and repetition is missing.
Section 2.5: The authors should describe method Purpad assay more specific. Solely referenced to paper is not scientific standard.
On the other hand, the conclusions are consistent with the arguments presented and the references are appropriate and most of them recent.
I have no comments on figures or tables.
